# Oxymatrine Alleviates Gentamicin-Induced Renal Injury in Rats

**DOI:** 10.3390/molecules27196209

**Published:** 2022-09-21

**Authors:** Songyao Kang, Tingting Chen, Zhihui Hao, Xiao Yang, Mingfa Wang, Zhifang Zhang, Sijia Hao, Fengting Lang, Hongxia Hao

**Affiliations:** 1Chinese Veterinary Medicine Innovation Center, College of Veterinary Medicine, China Agricultural University, Beijing 100193, China; 2Agricultural Biopharmaceutical Engineering Technology Research Center, Qingdao Agricultural University, Qingdao 266109, China; 3Qingdao University Medical Group Juxian Hospital, Qingdao 276599, China; 4College of Traditional Medicine, Inner Mongolia Medical University, Hohhot 010059, China

**Keywords:** gentamicin, nephrotoxicity, oxymatrine, oxidative stress, Nrf2/HO-1

## Abstract

Gentamicin is an aminoglycoside antibiotic commonly used to treat Gram-negative bacterial infections that possesses considerable nephrotoxicity. Oxymatrine is a phytochemical with the ability to counter gentamicin toxicity. We investigated the effects and protective mechanism of oxymatrine in rats. The experimental groups were as follows: Control, Oxymatrine only group (100 mg/kg/d), Gentamicin only group (100 mg/kg/d), Gentamicin (100 mg/kg/d) plus Oxymatrine (100 mg/kg/d) group (*n* = 10). All rats were treated for seven continuous days. The results indicated that oxymatrine alleviated gentamicin-induced kidney injury, and decreased rats’ kidney indices and NAG (*N*-acetyl-beta-d-glucosaminidase), BUN (blood urea nitrogen) and CRE (creatine) serum levels. The oxymatrine-treated group sustained less histological damage. Oxymatrine also relived gentamicin-induced oxidative and nitrative stress, indicated by the increased SOD (superoxidase dismutase), GSH (glutathione) and CAT (catalase) activities and decreased MDA (malondialdehyde), iNOS (inducible nitric oxide synthase) and NO (nitric oxide) levels. Caspase-9 and -3 activities were also decreased in the oxymatrine-treated group. Oxymatrine exhibited a potent anti-inflammatory effect on gentamicin-induced kidney injury, down-regulated the Bcl-2ax and NF-κB mRNAs, and upregulated Bcl-2, HO-1 and Nrf2 mRNAs in the kidney tissue. Our investigation revealed the renal protective effect of oxymatrine in gentamicin-induced kidney injury for the first time. The effect was achieved through activation of the Nrf2/HO-1 pathways. The study underlines the potential clinical application of oxymatrine as a renal protectant agent for gentamicin therapy.

## 1. Introduction

Gentamicin (GM) is an aminoglycoside antibiotic commonly used in treating bacterial infections, especially those caused by aerobic Gram-negative bacilli [1]. Its nephrotoxicity is a major concern from clinical perspective [2,3]. It has been estimated that over one-third of patients with long term gentamicin administration (more than 7 days) show signs of renal damage [4]. The renal toxicity of gentamicin contributes to its ability to activate apoptosis in renal tissue.

The main sites of gentamicin-induced nephrotoxicity are mesangial cells and renal proximal tubule cells [5,6]. Gentamicin can act on mitochondria and cause oxidative stress which leads to necrosis and apoptosis [7]. Gentamicin causes tubular injury featuring disappearance of the brush-like edge of the epithelial cells. Without intervention, tubular injury develops into acute tubular necrosis [8]. Gentamicin-induced renal toxicity can be evaluated by histopathologic and morphometric assays, and serum creatinine and blood urea nitrogen levels. Besides those long-established methods, *N*-acetyl-β-d-glucosaminidase (NAG) can also be used. It is a tubular lysosomal brush border enzyme, and is already widely used as a proximal tubular damage biomarker [9,10]. There are few effective pharmacological interventions to counter Gentamicin-induced nephrotoxicity; thus, it is critical to identify new agents to reduce the side effects of this drug.

Oxymatrine (OXY), C_15_H_24_N_2_O_2_, also known as Kushenin, is a phytochemical with a molecular weight of 264.369. It is an alkaloid extracted from the *Sophora flavescens*, which has been reported to display a wide range of pharmacological activities [11,12], such as anti-inflammatory, anti-oxidative and anti-apoptotic activities [13,14,15]. Reports have also suggested that oxymatrine has potent therapeutic effects on hepatitis B, rheumatoid arthritis, and mastitis in mice and rats [15]. Investigators found that oxymatrine reduced renal interstitial fibrosis and inflammation of obstructive renal lesions by inhibiting the release of a variety of inflammatory cytokines, including IL-6, IL-1β and TNF-α, as well as up-regulating the phosphorylated NF-κB p65 [16]. Meanwhile, oxymatrine has also been found to prevent liver injury via its anti-apoptotic and antioxidant activities. A previous study showed that oxymatrine could activate the Nrf2/HO-1 signaling pathway to protect against As_2_O_3_-induced oxidative damage [17]. Considering the above-mentioned properties of oxymatrine, it has the potential to reduce gentamicin nephrotoxicity in rats.

In our study, we investigated the renal protectant effect of gentamicin against gentamicin-induced nephrotoxicity in rats, and explored the possible mechanism of this effect.

## 2. Results

### 2.1. Oxymatrine Alleviated Gentamicin-Induced Nephrotoxicity in Rats

The organ coefficient of kidneys from the GM group was significantly higher than that in the control group (*p* < 0.01), as expected, which demonstrated gentamicin-induced nephrotoxicity. The oxymatrine-treated group had a lower organ coefficient compared to the GM group (*p* < 0.05) (Figure 1c). There was no statistically significant difference between the control and OXY groups. Rats from the GM plus OXY group showed less gross pathological changes in the kidney. NAG, CRE and BUN levels in the gentamicin-treated group were higher than in the control group (all *p* < 0.01). By contrast, the levels of NAG, CRE and BUN in rats in the OXY plus GM group markedly decreased to 640 μmol/L (*p* < 0.001) (Figure 1d), 28 mg/dL (*p* < 0.001) (Figure 1e) and 0.45 mg/dL (*p* < 0.05) (Figure 1f), respectively. There were no significant differences in NAG, BUN and CRE levels between the oxymatrine only group and the control group.

Histological evaluation revealed that oxymatrine treatment reduced kidney tissue damage caused by gentamicin in rats. In the GM group, the renal tissue damage was extensive. We observed degenerated glomeruli with enlargement of Bowman’s capsule, thickening of the capsule wall, the degeneration, dilation, and necrosis of renal tubules, and inflammatory cell infiltration (Figure 2c), which are all consistent with documented histological changes caused by gentamicin. Correspondingly, the histological scores reached 3.4 (*p* < 0.001) (Figure 1e), which was much higher than the control group. In the GM plus OXY group, the tubular necrosis and inflammatory cell infiltration was less severe (Figure 1d). The histological score from this group showed a marked decrease to 1.8 (*p* < 0.01) (Figure 1e). There were no obvious pathological changes in the kidneys from the OXY group and control group.

### 2.2. Oxymatrine Reduced Gentamicin-Induced Oxidative Stress in Renal Tissue

To determine the oxidative stress in renal tissue, we measured the biomarkers MDA, iNOS and NO and SOD, CAT and GSH, which are commonly used for this purpose. As shown in Table 1 MDA, iNOS and NO levels in GM treatment group significantly increased to 2.38 U/mg protein (*p* < 0.05), 1.81 U/mg protein (*p* < 0.05), and 891.8 mol/g protein (*p* < 0.001), respectively, and the SOD, CAT and GSH activities significantly decreased to 1025 U/mg protein, 68.8 U/mg protein, and 28.3 mmol/mg protein (all *p* < 0.05), respectively. Through these biomarkers, oxymatrine treatment showed a significant reduction of gentamicin-induced oxidative stress. We also found oxymatrine alone at 100 mg/kg had no effect on MDA, iNOS and NO levels and SOD, CAT and GSH activities, compared to the control group.

### 2.3. Oxymatrine Decreased Gentamicin-Induced Activation of Caspase-9 and Caspase-3 in Renal Tissues of Rats

After gentamicin administration at 100 mg/kg, the activities of caspase-9 and -3 increased 1.99- and 1.72-fold (both *p* < 0.001) compared to the control group, respectively. However, in the GM plus OXY group, these changes of caspase-9 and caspase-3 activities were less pronounced. Caspase-9 and caspase-3 activities decreased 1.23- and 1.15-fold (both *p* < 0.01) compared to the GM group, respectively (Figure 3a,b). The OXY group showed no changes in caspase-9 and caspase-3 activities compared to the control group.

### 2.4. Oxymatrine Surpressed Gentamicin Induced Inflammatory Mediators in Kidney

Gentamicin treatment significantly increased expression of the pro-inflammatory cytokines TNF-α, IL-6 and IL-1β (all *p* < 0.001), as expected. Oxymatrine co-administration suppressed this effect of gentamicin. In the GM plus OXY group, TNF-α, IL-6 and IL-1β levels decreased from 707.3, 62.66 and 159.4pg/mL protein to 628.6, 48.84 and 137.15 pg/mL protein (all *p* < 0.01) (Figure 2c–e).

### 2.5. Oxymatrine Down-Regulated the Bax, NF-κB mRNAs Expression and Up-Regulated the Expression of Bcl-2, Nrf2 and HO-1 mRNAs

Gentamicin treatment significantly up-regulated Bax, Nrf2, HO-1 and NF-κB mRNA expression (all *p* < 0.01) (Figure 4b–e), but not Bcl-2 mRNA (*p* < 0.001) (Figure 4a). Oxymatrine down-regulated (*p* < 0.05) Bax and NF-κB mRNAs expression, increased Bcl-2, Nrf2 and HO-1 mRNA expression (all *p* < 0.01), in contrast to the gentamicin group (Figure 4). The expression of Bax and NF-κB mRNAs was not affected in the control group or the OXY group. In the OXY group, Bcl-2, Nrf2 and HO-1 mRNA expression was up-regulated (*p* < 0.05).

## 3. Discussion

Gentamicin is an effective aminoglycoside antibiotic for the treatment of bacterial infections. The major side effect of this drug is renal toxicity, which is characterized by tubular necrosis, thus limiting its clinical application [18,19]. The exact mechanism of gentamicin-induced nephrotoxicity is still under investigation. From a clinical perspective, it is reasonable and necessary to find a way to alleviate the adverse effects of gentamicin without reducing its therapeutic benefits.

Previous researchers have speculated that oxidative stress and ROS are the key factors leading to tubular necrosis and the decrease of glomerular filtration rate [3,18,20]. Consistent with these findings, we found higher levels of SOD and CAT and lower levels of GSH in the kidneys of the gentamicin-treated group, which indicated that gentamicin decreased the tissue total antioxidant capacity and elevated the lipid peroxide (MDA), iNOS and NO levels. It is logical to assume that increasing the tissue antioxidant capacity could prevent or at least decrease the severity of renal tissue damage caused by gentamicin. In our investigation, oxymatrine administration increased SOD and CAT activity. Nuclear respiratory factor 2 (Nrf-2), an important transcriptional factor that can reduce oxidative stress by increasing antioxidant defense, was also indicated [21,22]. Oxymatrine administration upregulated the expression of Nrf2 and HO-1 mRNA. The antioxidative effect of oxymatrine is not specific to the kidney or gentamicin. In our previous work, we found that oxymatrine prevented the accumulation of MDA in hypoxic-ischemic-induced oxidative stress in neonatal rats, and enhanced antioxidant enzymes activities (SOD, T-AOC, CAT, GSH-PX) at the same time, suggesting that oxymatrine reduced the severity of HIBD in neonatal rats by decreasing lipid peroxide and improving the antioxidant defense system [23]. Other researchers have also reported that oxymatrine reduces doxorubicin-induced myocardial tissue damage and improves liver function [22,24,25] via inhibition of oxidative activity. The mechanism of oxymatrine’s antioxidative effect is most likely through radical scavenging activity, and the ability to activate intrinsic antioxidant defense.

Inflammation is another important component of gentamicin-induced nephrotoxicity [26,27]. The NF-κB signaling pathway regulates the expression levels of many genes related to inflammation; therefore, it is considered to be a vital transcription factor that can mediate acute and chronic inflammation by transcription of genes encoding cytokines, chemokines, adhesion molecules, pro-inflammatory enzymes and apoptosis-regulating proteins, leading to apoptosis and interstitial fibrosis in progressive renal disease [28,29]. In the current study, gentamicin-induced inflammatory responses in renal tissues up-regulated the expression of NF-κB and promoted the production of TNF-α, IL-1β and IL-6, resulting in inflammatory cell infiltration [30,31]. Oxymatrine down-regulated NF-κB expression and significantly decreased pro-inflammatory cytokines like TNF-α, IL-1β and IL-6. Oxymatrine also suppressed the production of NF-κB, TNF-α, IL-6, and ICAM-1.

Our study showed that oxymatrine is a promising candidate as an adjuvant drug in clinical rodent colitis therapy [32]. The factors limiting oxymatrine’s clinical application include its short half-life, low bioavailability, and whole-body distribution. New formulations and drug delivery systems can improve the bioavailability of oxymatrine. HSPC liposomes and oxymatrine-loaded nanostructured lipid carriers have been made to improve anti-inflammatory activity of oxymatrine, which could prolong oxymatrine retention time and obtain high therapeutic levels in the liver [33,34].

We found that oxymatrine alleviated gentamicin-induced nephrotoxicity by reducing oxidative stress and inflammatory response in rats. Specifically, oxymatrine inhibited intracellular ROS formation and activated the Nrf2/HO-1 antioxidant signaling pathway to reduce cell apoptosis. Oxymatrine also reduced inflammation by inhibiting the NF-κB signaling pathway (Figure 5). Our study indicates that oxymatrine can be a potential compound for reducing gentamicin toxicity in human and veterinary clinical applications.

## 4. Materials and Methods

### 4.1. Chemicals

Gentamycin Sulfate Injection (2 mL: 80,000 U) was purchased from Shandong lukang pharmaceutical Co., Ltd. (Jining, China). Oxymatrine (≥98%) was obtained from Damas-beta Reagent Co., Ltd. (Shanghai, China). All other chemicals were analytical grade.

### 4.2. Animals

Forty Wistar rats (male, 250–300 g) purchased from Pengyue laboratory Animal Breeding Co., Ltd. (Jinan, China) were used. The housing conditions were as follows: 23 ± 2 °C, 60 ± 5% relative humidity, a 12 h light/dark cycle. All the rats were given a 1 week acclimatization period before the experiment, and had adequate water and feed all the time. All the animal experiments were approved by the Committee of Animal Use and Protection of China Agriculture University (AW92502202-2-1).

### 4.3. Experimental Design

Forty rats were randomly assigned into four groups: Control, OXY group, GM group, GM plus OXY group (*n* = 10). All the testing substances except oxymatrine were dissolved in 1 mL 0.5% CMC-Na solution, and administrated by intraperitoneal injection. Oxymatrine was dissolved in 1 mL 0.5% CMC-Na solution and administrated by gavage. All rats were treated once per day for 7 consecutive days [16]. In the control group, rats were injected 1 mL 0.5% CMC-Na solution only. In the OXY group, rats received oxymatrine at a rate of 100 mg/kg. In the GM group, rats received gentamicin at a rate of 100 mg/kg. In the GM plus OXY group, rats received oxymatrin at a rate of 100 mg/kg in 1 mL solution 1 h prior to gentamicin injection at 100 mg/kg.

Twelve hours after last treatment, blood samples were collected from each rat. The rats were euthanized with sodium pentobarbital (80 mg/kg, i.p). Afterwards, rat kidneys were collected to determine the weight and kidney organ coefficient (organ coefficient = organ weight/body weight), and stored at −80 °C for further analysis.

#### 4.3.1. Measuring Blood Urea Nitrogen (BUN) and Serum Creatinine (s-CRE) Levels

The BUN and CRE levels in serum were determined through Automatic Analyzer (IDEXX Catalyst One^®^) using the standard diagnostic kits (IDEXX biological products trading Co., Ltd., Shanghai, China).

#### 4.3.2. Measuring SOD, GSH, CAT Activities and MDA, NO and iNOS Levels

Kidney tissues were homogenized in 9 parts (9:1 *v*/*w*) of cold 50 mM Tris buffer (pH 7.4) with a mechanical grinder. We collected the supernatant after centrifugation (4000× *g*, 15 min) at 4 °C. SOD, GSH, CAT activities, MDA, NO, and iNOS concentration in the supernatants were measured with commercial testing kits (Nanjing Jiancheng, Nanjing, China). The concentration of all proteins was determined by BCA protein assay kits (Beyotime Biotechnology, Haimen, China).

#### 4.3.3. Histopathological Examination

We randomly selected the kidneys of 5 rats from each group. The specimens were fixed in 10% formalin for 48 h, dehydrated in alcohols of different concentrations and xylene transparent, then embedded with paraffin, sectioned into 5 μm slices. These slices were dewaxed and stained by hematoxylin–eosin (H&E) for optical microscopic examination [12]. To evaluate the degree of kidney injury in different groups, the histopathological changes in tubular epithelial alterations were examined, such as desquamation, vacuolization, casts, tubular dilation, and inflammatory cell infiltration, and the severity of lesions for renal tissues was evaluated by a semi-quantitative score (Grade 0–5) [35]. The scoring system is presented in Table 2. Pictures of the samples (50 μm) were taken under a light microscope CX41(Olympus, Hamburg, Germany) at a visual magnification of 20×.

#### 4.3.4. Measurement of the Caspase-9 and Caspase-3 Activities and the IL-6, IL-1β and TNF-α Levels

The levels of caspase-9, caspase-3, IL-6, IL-1β and TGF-α in the renal tissue were determined using ELISA kits as directed by the manufacturer (Jiangsu Jingmei Biotechnology Co., Ltd., Shenzhen, China).

#### 4.3.5. Quantitative Reverse-Transcription PCR

Total RNA from 10 mg of each frozen kidney sample was extracted by RNeasy Mini Kit following kit protocols (Life Technologies, Grand Island, NY, USA). The OD of RNA at 260/280 nm was determined for quality control. Approximately 1 μg RNA from each tissue sample was employed to synthesize the cDNA by using the Prime Script RT-PCR kit (Takara, Dalian, China). The PCR primers are presented in Table 3. RT-PCR tests were processed using the ABI QuantStudio™7 detection system. Target gene expression was normalized to GAPDH, and the (2^−ΔΔ^^CT^) method was used for the calculation of fold changes in gene expression [36] All reactions were run in triplicate.

PCR reactions were conducted under the conditions described in Table 4. RT-PCR tests were processed using the ABI QuantStudio™7 detection system. Target gene expression was normalized to GAPDH, and the (2^−ΔΔ^^CT^) method was used for the calculation of fold changes in gene expression [36]. All reactions were run in triplicate.

### 4.4. Statistical Analyses

All results are presented as mean ± SD (*n* = 10), and statistical analyses were conducted using GraphPad Prism 8.0 (GraphPad Software, San Diego, CA, USA), and comparisons between groups were performed via one-way ANOVA, whereas multiple comparisons were performed using the LSD method. *p* < 0.05 was defined as being statistically significant.

## Figures and Tables

**Figure 1 molecules-27-06209-f001:**
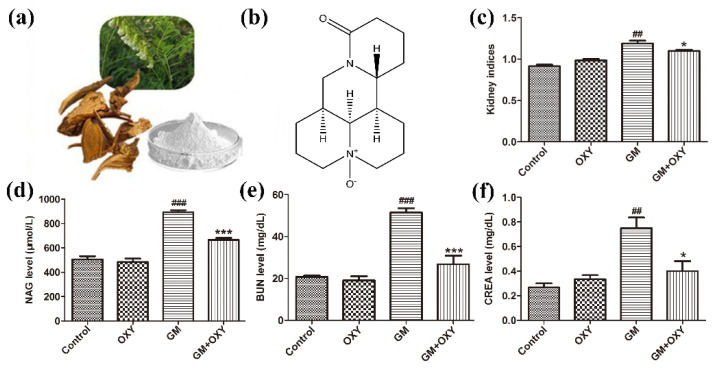
Oxymatrine alleviated gentamicin-induced nephrotoxicity in rats. (**a**) Sophora flavescens ait (the Chinese herb Kushen). (**b**) Chemical structure of OXY; (**c**) kidney indices (kidney organ coefficient). (**d**) Urinary NAG levels. (**e**) Serum BUN and (**f**) CRE levels. Data are presented as means ± SD (*n* = 10). ^##^ *p* < 0.01 and ^###^ *p* < 0.001 compared to the control group; * *p* < 0.05 and *** *p* < 0.001 compared to the GM treatment group.

**Figure 2 molecules-27-06209-f002:**
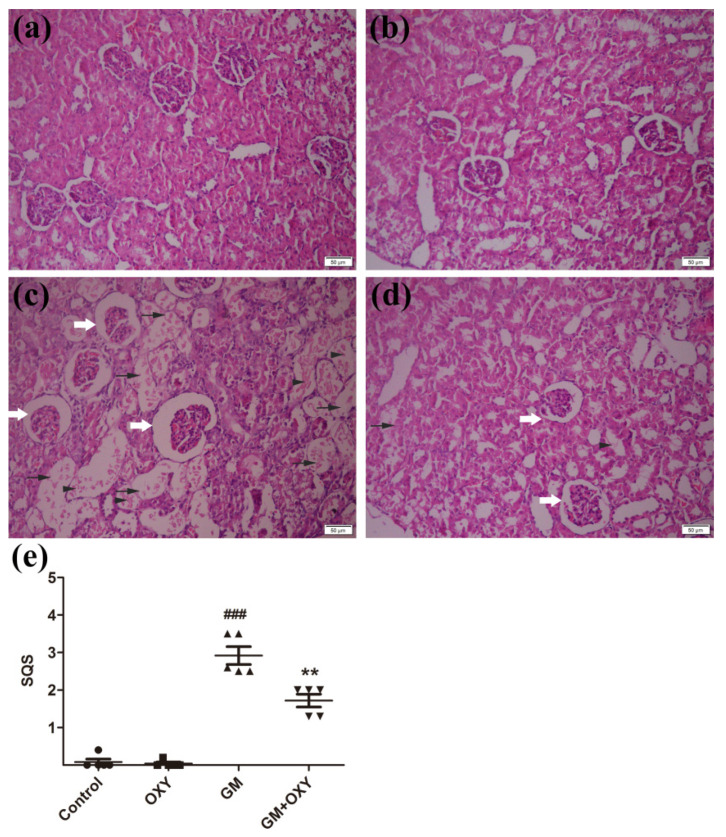
(**a**) Control group: no significant damage. (**b**) Oxymatrine group: no significant damage. (**c**) Gentamicin group: marked tubular damage with necrosis and exfoliation of epithelial cells (arrows), cast formation (arrowheads) and glomerular abnormalities (open arrow). (**d**) Gentamicin + Oxymatrine group: minor tubular with necrosis (arrows) of epithelial cells, cast formations (arrowheads) and glomerular abnormalities (open arrow). (**e**) Semiquantitative scores of kidney damage (group the means ± SD, *n* = 5). ▲ indicated specimens, ^###^ indicated. The scores are significantly higher in GM + OXY, *p* < 0.001 compared to the control group (indicated as ^###^) and significantly lower than GM group (indicated as **) *p* < 0.01 and *p* < 0.001. OXY inhibited gentamicin-induced renal inflammatory responses.

**Figure 3 molecules-27-06209-f003:**
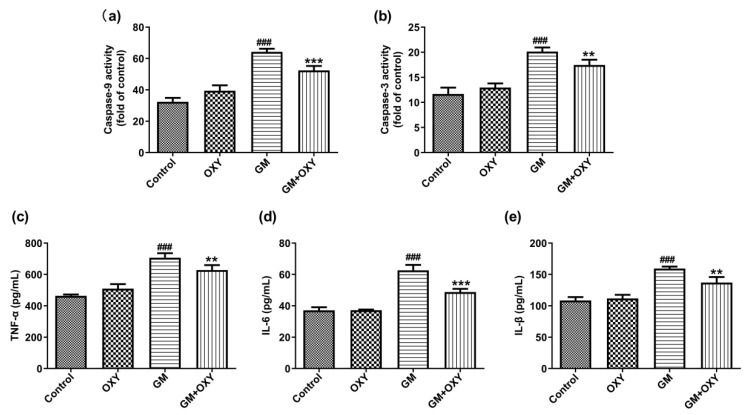
Effect of oxymatrine on the apoptosis markers caspase-9 (**a**), caspase-3 (**b**) and pro-inflammatory markers TNF-α (**c**), IL-6 (**d**) and IL-1β (**e**) in the serum of rats treated with gentamicin. ELISA results are presented as the mean ± SD (*n* = 10). ^###^ *p* < 0.001 compared to the control group; ** *p* < 0.01 and *** *p* < 0.001compared to the solely GM treatment group.

**Figure 4 molecules-27-06209-f004:**
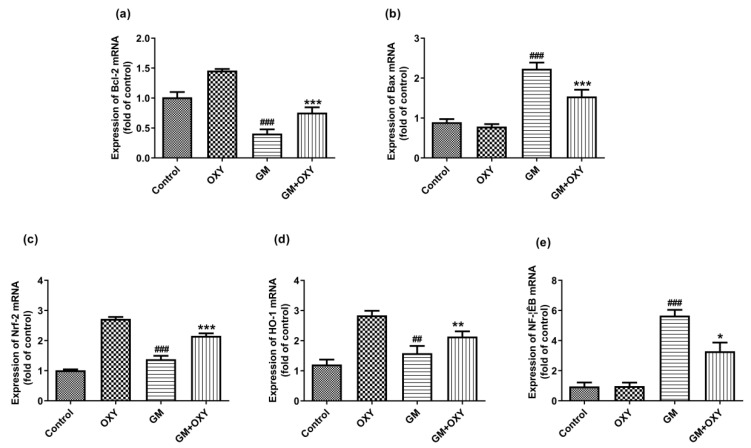
Effect of oxymatrine on GM-induced expression of Bcl-2 (**a**), Bax (**b**), Nrf-2 (**c**), HO-1 (**d**) and NF-κB (**e**) mRNAs in the kidneys. Data are presented as mean ± SD (*n* = 10). * *p* < 0.05, ** *p* < 0.01 and *** *p* < 0.001 compared to the control group; ^##^ *p* < 0.01 and ^###^ *p* <0.001, compared to the GM treatment group.

**Figure 5 molecules-27-06209-f005:**
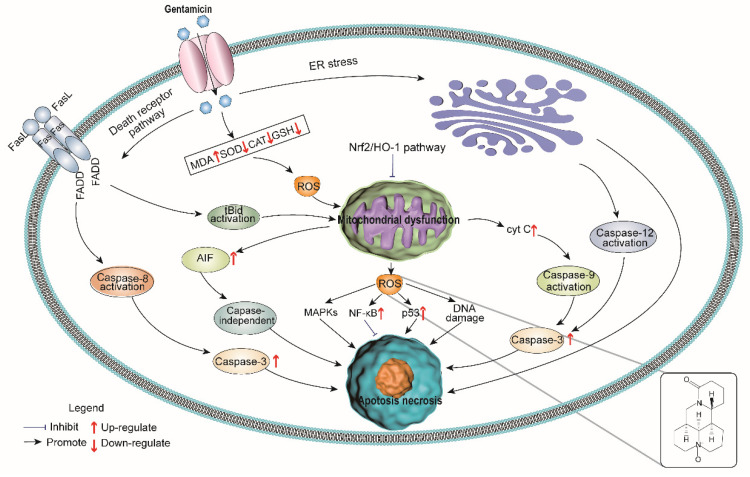
Model of oxymatrine renoprotection against gentamicin-induced nephrotoxicity.

**Table 1 molecules-27-06209-t001:** Effect of oxymatrine pre-treatment on the levels of oxidative and nitrative stress markers in the renal tissues of rats treated with gentamicin.

Biomarker	Treatment Group			
	Control Group	OXY Group	GM Group	GM plus OXY Group
MDA (mmol/mg of protein)	1.58 ± 0.069	1.68 ± 0.011	2.38 ± 0.092 ^#^	1.92 ± 0.20 *
SOD (U/mg of protein)	1106 ± 1.414	1081 ± 6.899	1025 ± 6.980 ^##^	1145 ± 47.13 *
CAT (U/mg of protein)	98.8 ± 4.76	95.2 ± 3.40	68.8 ± 6.74 ^#^	96.1 ± 2.89 *
GSH (mmol/mg of protein)	36.7 ± 2.80	36.4 ± 2.48	28.3 ± 2.23 ^#^	36.6 ± 1.97 **
iNOS (U/mg of protein)	0.811 ± 0.027	0.764 ± 0.028	1.81 ± 0.21 ^#^	0.845 ± 0.09 **
NO (mol/g of protein)	488.1 ± 39.86	483.4 ± 40.07	891.8 ± 30.28 ^###^	665.6 ± 37.64 ***

Data are presented as mean ± SD (*n* = 10). ^#^
*p* < 0.05, ^##^
*p* < 0.01 and ^###^
*p* < 0.001, OXY group or GM group vs control group; * *p* < 0.05, ** *p* < 0.01 and *** *p* < 0.001, GM plus OXY groups vs GM group.

**Table 2 molecules-27-06209-t002:** The scoring system for each kidney sample.

Scores	Severity of Lesions
0	no pathological change
+1	mild change
+2	mild to moderate change
+3	moderate change
+4	moderate to severe change
+5	severe pathological change

**Table 3 molecules-27-06209-t003:** The sequences of the primers (5′−3′) used for qRT-PCR.

Gene	Primer Sequence (5′−3′)
Nrf2	5′-CAC ATT CCC AAA CAA GAT GC-3′ 5′-TCT TTT TCC AGC GAG GAG AT-3′
HO-1	5′-CGT GCT CGA ATG AAC ACT CT-3′ 5′-GGA AGC TGA GAG TGA GGA CC-3′
NF-κB	5′-CAC TGT CTG CCT CTC TCG TCT-3′ 5′-AAG GAT GTC TCC ACA CCA CTG-3′
Bax	5′-CCA AGA AGC TGA GCG AGT GTC-3′ 5′-TGA GGA CTC CAG CCA CAA AGA-3′
Bcl-2	5′-CCG GGA GAT CGT GAT GAA GT-3′ 5′- ATC CCA GCC TCC GTT ATC CT-3′
GAPDH	5′-ACA GTC CAT GCC ATC ACT GCC-3′ 5′-GCC TGC TTC ACC ACC TTC TTG-3′

**Table 4 molecules-27-06209-t004:** PCR reaction program.

Step	Temperature	Time	Instructions
1	95 °C	5 min	initial activation
2	95 °C	30 s, 40 cycles	denaturing
3	60 °C	30 s	annealing
4	72	30 s	elongation.

## Data Availability

Not applicable.

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
