# Peer review of "Oxymatrine Alleviates Gentamicin-Induced Renal Injury in Rats"

_molecules, 2022, doi:10.3390/molecules27196209_

Round 1

Reviewer 1 Report

The paper shows the nephroprotective effect of oxymatrine on gentaminice-induced kidney injury in rats.

Overall, the work was well conducted has a valid premiss. However, some adjustments must be done before the paper can be accepted for publication:

  • Why did the authors choose to administrate OXY by i.p. route? As published previously, OXY could be administrated by gavage. This route of administration is more suitable, considering the possibility of this substance for human use.
  • Also, it is unclear why OXY and GM were diluted in 0.5% CMC solution. 
  • In lines 255-256, it is stated that "rats kidneys were collected to determine the weight and kidney indices". What exactly are these kidney indices? This must be clarified.
  • In section 4.3.5 (Quantitative reverse-transcription PCR), the paper does not describe the transcriptase reverse method. in this way, it is not clear how the RNA samples extracted were used in qPCR analysis.
  • In section 4.4, I believe there was a mistake regarding the significance criteria selected. Please correct it.
  • The histopathological results presented by the authors are difficult to evaluate, since there are no arrows or indications of the alterations cited by them. I recommend that these indications must be included in the figure. Also, the groups treated only with OXY also presents some of the features described for the GM group (as such as enlargement of Bowman’s capsule). This whole figure must be thoroughly revised.
  • The results for caspase activity are not expressed as fold of control, as stated by the authors. This must be corrected.
  • The discussion section is constructed very poorly. The authors simply repeated the description of the results. This whole section must be rewritten, focusing on the comparison of the results with the literature and adequate cientific inference.

Author Response

Dear reviewer,

Thank you for reviewing our paper. We appreciate your help very much. We have revised the manuscript according to your suggestions.

  1. Oxymatrine was administrated by gavage. The original description was not clear enough, we have changed the wording to make it more precise.
  2. GM and OXY was deluted in 0.5% CMC solution because solubility of Oxymatrine in pure water was low. CMC solution helps oxymatrine destribute more evenly in the solution.  To limit the variable, we also delute GM in CMC solution instead of  injection water.
  3. Kidney indice is not the proper word. We have changed it to kidney organ coeffecient. Kidney organ coeffecient= kidney weight/body weight.
  4. We have added more detailed description regarding qPCR method.
  5. We have corrected the significance criteria in section 4.4
  6. We have added indication in supplementary figure as you suggested. We also had a discussion regarding OXY figure, most of us tend to think it was an incidental finding, and may not necessarily related to the oxymatrine treatment.
  7. The figure wil be corrected.
  8. The discussion section has be re-orgnised to make it more concise.

Reviewer 2 Report

Review of the manuscript entitled: Oxymatrine Alleviates Gentamicin-Induced Renal Injury in Rats. The manuscript is interesting but some corrections will be needed. The manuscript is interesting but some corrections will be needed. Please check the entire manuscript carefully e.g. lines 38, 43, 45, 69, 82 and so on

It is very important to present the purpose of the manuscript. The chapter introduction should end like this “The aim of this study is to evaluate/or/study …”

Line 95-98 figure should be corrected, what number is it? no description what (a) means and so on? If you are showing a figure there should be a description.

The description of the results is correct. The discussion is concrete and correct, I believe it is supported by the results.

4. Materials and Methods

4.1. Chemicals - It is impossible that the authors used only two chemical reagents. Please list more key reagents.

Lines 262-268 - commercial testing kits catalog numbers should be provided.

Line 282-286 – similar commercial ELISA kits catalog numbers should be provided.

Author Response

Dear reviewer,

Thank you for reviewing our manuscript. We appreciate your effort to help us improve this article. We have revised the manuscript according to reviewers suggestion. The items you have mentioned will be explained below.

  1. We have checked the manuscript and removed the extra punctuations.
  2. We have added the study objectives in the intruduction section.
  3. We have added the figure description for supplementary figure.
  4. Only two  substances were used in animal experiment. Other chemicals either come in the form of commercial kits or described in experiment design. 
  5. We have tried to track down kit catalog numbers, but unfuturnately it wasn't available. And due  to the Covid epidemic further searching may not be possible.

Reviewer 3 Report

This interesting study suggests that treatment of Oxymatrine Alleviates Gentamicin-Induced Renal Injury in 2 Rats. This may be accepted after major revision.

1. In the abstract, all the abbreviated words should be defined in complete forms for example NAG, BUN, etc.

2. The objective of the study must be explained in the introduction.

3. In Figure 3 e, why GM treatment is showing better expression of NF expression than GM+OXY, please explain.

4. Please add supplementary figure e in the main to better understand the morphological changes due to treatments.  

5. Why only male rats were considered in the study, any specific reason. 

Author Response

Dear reviewer,

Thank you for reviewing our manuscript. We appreciated your effort to help us improve this article. We have made adjustment according to your suggestions. 

1.The complet forms has been added to corresponding abbreviation.

2. We have added the objective of this study in the introduction section.

3. GM increased NF-kB mRNA expression is a part of the GM induced injury. NF-kB is an key factor mediate acute and chronic inflammation. In healthy tissue the expression of this factor remains low. In the mentioned figure, the low expression of NF-kB mRNA in control and OXY groups indicated the abcent of renal injury/inflammation. The highest expression in GEM group reflected the most severe tissue damage. GEM+OXY group is higer than control and and OXY groups, but lower than GEM group, suggested oxymatrine can not completely prevent the tissue damage/inflammation, but can somehow alleviated it.

4. We will adjust the figure.

5. We use only male rats to reduce the number of variables. We may explore oxymatrine's effect in animals of different sex in the future.

Round 2

Reviewer 1 Report

The authors corrected the manuscript accordingly.

Reviewer 2 Report

The authors corrected the manuscript according to my suggestions

Reviewer 3 Report

The revised manuscript has improved and most of the issues have been addressed. Considering this, the revised manuscript may be now accepted for publication.